# OpenReview forum: "MISR: Measuring Instrumental Self-Reasoning in Frontier Models"
_ICLR.cc/2025/Conference — Submitted to ICLR 2025_

### Official Review · Reviewer_wZBb · 2024-10-25

**Soundness:** 3
**Presentation:** 2
**Contribution:** 2
**Rating:** 3
**Confidence:** 4

**Summary:**

This work evaluates the instrumental self-reasoning abilities of several mainstream cutting-edge LLMs in agentic scenarios through five different perspectives on a task that needs to reveal implicit goals, and conducts large-scale experimental analysis. The experimental results show that only the most advanced current LLMs demonstrate some performance in this ability, and none of the LLMs passed the most difficult tasks. The authors believe that these results can draw attention to this capability of LLMs, thereby contributing to AI safety and development.

**Strengths:**

The work designed a large number of tasks and conducted extensive experiments, the results of which seem to be highly credible and reproducible. At the same time, the author organized these tasks into a benchmark that can be used to evaluate the instrumental self-reasoning ability of subsequent LLMs.

Additionally, researching the instrumental self-reasoning ability of LLMs is a very interesting issue, as it can demonstrate whether LLMs are capable of adjusting their settings through reasoning to perform different tasks in a targeted manner.

**Weaknesses:**

In terms of writing, the overall structure of this work resembles a technical report. It lacks the completeness and rigor expected of an academic work and does not clearly demonstrate its positioning within the current field or the significance of the research. For instance, it does not highlight the advantages of instrumental self-reasoning compared to self-correction [1] or self-improvement through introspection [2].

In terms of experiments, this work lacks detailed experimental setup information, including task quantity, task sources, and the motivation behind designing these tasks. Additionally, the paper lacks details on the settings for the LLM, such as temperature and seed.

In terms of novelty, I do not see much improvement compared to the instrumental self-reasoning evaluation part of the work [3] cited by the author themselves (the specific experimental task format design as well). The author also mentioned that their focus is on agentic scenarios, which were lacking in previous work. However, I do not believe that [3] lacks this aspect. In fact, this capability itself needs to be demonstrated in agentic scenarios.



**Reference**

[1] Large Language Models Cannot Self-Correct Reasoning Yet (https://arxiv.org/abs/2310.01798)

[2] Recursive Introspection: Teaching LLM Agents How to Self-Improve (https://openreview.net/forum?id=UPoQqreegH)

[3] Evaluating Frontier Models for Dangerous Capabilities (https://arxiv.org/abs/2403.13793)

**Questions:**

See the **Weaknesses** section.

---

> ### Author Response · Authors · 2024-11-17
>
> Thank you for recognizing the credibility and reproducibility of our results, and for highlighting the importance of studying instrumental self-reasoning in LLMs.
>
> We appreciate your suggestions regarding related work on self-correction and self-improvement. While these are important and related capabilities, self-reasoning as we define it is distinct: self-correction refers to an agent’s ability to recognize and fix mistakes, while self-improvement focuses on enhancing its own capabilities. In contrast, self-reasoning encompasses “the ability of an agent to deduce and apply information about itself in service of an objective.” This broader definition includes but extends beyond self-correction and self-improvement to include understanding one’s role and capabilities within an environment. We will expand our literature review to better articulate these distinctions.
>
> We would appreciate if you could elaborate on your comment about the paper resembling a technical report. We agree that the discussion of related work can be improved, and we will include a more detailed discussion of related literature on self-correction and self-improvement. Are there other parts of the paper that you think are lacking and are too similar to a tech report?
>
> > In terms of experiments, this work lacks detailed experimental setup information, including task quantity, task sources, and the motivation behind designing these tasks. Additionally, the paper lacks details on the settings for the LLM, such as temperature and seed.
>
> Regarding experimental details: In appendix C, you can find full descriptions of each of the tasks and their variations. Due to the length of these descriptions, we do not include them in the main body of the paper. We appreciate the reminder to include LLM generation settings, which we will include in the appendix in our revision. Additionally, as mentioned in the paper, we plan to open-source our complete implementation using the reproducible Inspect framework for the camera-ready version.
>
> > In terms of novelty, I do not see much improvement compared to the instrumental self-reasoning evaluation part of the work [3] cited by the author themselves (the specific experimental task format design as well).
>
> Regarding novelty and positioning within the field: We address this extensively in our response to all reviewers above, but in brief, while Phuong et al. provided valuable groundwork with 3 evaluation tasks, we contribute 14 novel tasks across diverse categories of embedded self-reasoning, representing a substantial expansion in both scope and depth of evaluation.
>
> > The author also mentioned that their focus is on agentic scenarios, which were lacking in previous work. However, I do not believe that [3] lacks this aspect. In fact, this capability itself needs to be demonstrated in agentic scenarios.
>
> Thank you for pointing this out. We were referring specifically to the SAD dataset proposed by Laine et al. (the most relevant and large scale benchmark to ours), not Phuong et al. We will make this distinction more clear in our revisions.

---

> > ### Comment · Reviewer_wZBb · 2024-11-20
> >
> > Thank you for the explanation. I understand the definition of self-reasoning in this paper, and it is indeed different from self-correction or self-improvement. Although the authors have demonstrated several differences compared to Phuong et al. 2024, I believe that merely increasing the number of tasks or introducing additional settings is not sufficient. I still do not see enough novelty and contribution from this work.
> >
> > The authors hope me to point out the parts of the paper that resemble a technical report. Here are a few examples. First, most of the paper focuses on task definitions and technical details while neglecting in-depth consideration of the research question itself and what new insights it brings to the related field. This might be one of the reasons why reviewers question the novelty. Secondly, there are too many scattered and brief subsections (such as sections 3.1-3.6). In my opinion, these parts should not take up such a large portion of the paper and should be briefly described using paragraphs. As a research paper, more space should be devoted to introducing the problem the research aims to solve, the core methodology used, and the underlying motivation, as well as demonstrating the rigor of the experiments from multiple perspectives. Lastly, the evaluation did not point out some decisive issues, such as what conclusion these evaluation results bring to future research on LLMs.

---

> > > ### Author Response · Authors · 2024-11-22
> > >
> > > Thank you for your detailed feedback regarding the paper's structure and novelty. We appreciate the follow up. However, we would like to push back on a couple of points.
> > >
> > > First, regarding novelty and scientific contribution, our work goes well beyond simply adding more tasks/settings:
> > >
> > > 1. We provide qualitatively different evaluations that capture distinct aspects of embedded self-reasoning. While Phuong et al. focused primarily on self-modification, our framework evaluates different aspects of how agents understand and operate within their environment. These more diverse evaluations help to gain a better scientific understanding of what constitutes self-reasoning in AI systems beyond the pure notion of self-reasoning as a dangerous capability in Phuong et al
> > > 2. We are the first to conduct systematic evaluations across different models for self-reasoning capabilities. This empirical contribution provides insights into how these capabilities develop across different frontier models -- a key scientific question that previous work has not addressed.
> > > 3. Our monitoring experiments are the first study of models’ ability to evade self-reasoning detection. This aspect has not been considered in prior work at all.
> > >
> > > Regarding the paper’s structure and focus, we note that other reviewers actually requested more detailed task descriptions in the main paper. We will revise the paper to highlight the scientific contributions discussed above and make it feel less like a tech report. However, we must balance providing sufficient technical detail for reproducibility while maintaining focus on broader implications.

---

### Official Review · Reviewer_oVYg · 2024-10-29

**Soundness:** 2
**Presentation:** 3
**Contribution:** 2
**Rating:** 5
**Confidence:** 3

**Summary:**

This paper proposes a suite of tasks to evaluate potentially dangerous self-reasoning capabilities in frontier models. These tasks measure capabilities such as self modification, tool improvement and knowledge seeking. Their work builds on the UK AI Safety Institute’s “Inspect” framework, and involves having an LLM agent execute bash commands in a container to solve tasks with implicit self-reasoning challenges.

**Strengths:**

1. This work is highly relevant, and research along these lines will be crucially important as frontier models continue to progress. The authors have identified an important area to investigate and are thinking along good lines.
2. The writing is concise and informative.
3. I like the thorough results in appendix, and a detailed run-through of the prompts used is helpful for understanding exactly what's going on.

**Weaknesses:**

1. I am concerned as to how novel this work is given Phuong et al’s “Evaluating Frontier Models for Dangerous Capabilities”. The “self modification” task is not novel, whilst “tool improvement” is better but seems slightly unnecessary given the suite of tasks in Phuong et al’s “self-proliferation” section. I would appreciate a more detailed comparison between your tasks and theirs, particularly for the "tool improvement" tasks, explaining how you contribute beyond existing evaluations.
2. The relevance of the “knowledge seeking” task to instrumental self-reasoning seems slightly tenuous. The prompts ask models to explore the environment, and the dangerous behavior is then whether they do explore the environment? Could you please clarify how the knowledge seeking tasks specifically evaluate instrumental self-reasoning rather than simple instruction following.
3. I would like to see confidence intervals on the bar charts.
4. I think that “our evaluations significantly improve upon prior work” and “our results have strong implications for AI safety and development” are overclaims here. I would like to hear a defense as to how this work “significantly” improves upon the highly detailed Phuong paper mentioned above.

**Questions:**

Please respond to the weaknesses highlighted above.

---

> ### Author Response · Authors · 2024-11-17
>
> Thank you for recognizing the importance of this research direction and the quality of our documentation.
>
> Regarding the comparison to Phuong et al.’s work: We want to clarify that our tool improvement tasks are entirely novel and distinct from the self-proliferation tasks in their paper. While both works examine aspects of agent capabilities, they focus on different phenomena. Our tool improvement tasks specifically evaluate an agent’s ability to reason about and modify its extended capabilities as part of embedded self-reasoning (as detailed in our general response). The self-proliferation tasks in Phuong et al. seek to measure whether agents can act autonomously at all. Additionally, while Phuong et al. provide valuable groundwork on self-reasoning evaluations, they only provide 3 evaluation tasks focused primarily on self-modification. We contribute 14 novel tasks across multiple diverse categories of embedded self-reasoning, offering a more comprehensive evaluation framework for self-reasoning. We provide a longer discussion of our paper’s novelty in our general comment above.
>
> Regarding knowledge seeking: These tasks are fundamental to embedded self-reasoning because they evaluate whether an agent can reason about its own knowledge state relative to its environment. Just as humans must understand what they do and don't know to effectively navigate their environment, an agent must be able to reason about its own knowledge gaps to effectively operate as an embedded agent. Our general response includes more discussion of how our tasks connect to the broader framework of embedded self-reasoning.
>
> We will add confidence intervals to all bar charts in the revision to provide better statistical context for our results. Thank you for the suggestion.
>
> Regarding potential overclaiming: We acknowledge that the phrasing “our results have strong implications for AI safety and development” may overstate the immediate impact of our findings and we will remove this sentence from the paper. However, we stand by our claim of substantial improvements to prior work in this specific area -- while Phuong et al. first identified instrumental self-reasoning as an important capability to measure, our work provides the first comprehensive evaluation framework specifically focused on embedded self-reasoning, with a significantly larger and more diverse set of tasks. We believe this represents a meaningful step forward in our ability to measure and understand these capabilities.

---

> > ### Comment · Reviewer_oVYg · 2024-11-23
> >
> > Thank you for your response. Just to clarify, my review stated "the 'self modification' task is not novel", and your response is to point out that the "tool improvement" tasks are novel - not addressing the concern I raised. You then go on to claim that all 14 tasks are novel, without evidencing this. I'm a little confused as to how this responses addresses my concerns, apologies if I am misreading.

---

> > > ### Author Response · Authors · 2024-11-23
> > >
> > > Thank you for the clarification. We apologize for not directly addressing your concern about self-modification tasks in our previous response.
> > >
> > > You are correct that 2 of our 4 self-modification tasks build directly on those in Phuong et al. We should have acknowledged this explicitly in our individual response rather than only referencing our general comment to all reviewers. To briefly summarize our general comment here, our contribution includes:
> > >  - 2 novel self-modification tasks beyond those in Phuong et al.
> > >  - 12 entirely novel tasks across 4 other categories of self-reasoning
> > >  - The first systematic cross-model comparison of self-reaoning
> > >  - Novel monitoring experiments revealing models' ability to conceal their self-reasoning
> > >
> > > We focused our previous response on tool improvement because you specifically mentioned it might be "unnecessary given the suite of tasks in Phuong et al's 'self-proliferation' section." We wanted to emphasize that our tool improvement tasks evaluate fundamentally different capabilities than the self-proliferation tasks in their work.
> > >
> > > We hope this clarifies both the novelty of our individual tasks and how they differ from prior work. Please let us know if you would like additional details about any specific tasks or categories.

---

### Official Review · Reviewer_byfs · 2024-11-03

**Soundness:** 3
**Presentation:** 3
**Contribution:** 3
**Rating:** 8
**Confidence:** 3

**Summary:**

This paper introduces MISR (Measuring Instrumental Self-Reasoning in Frontier Models), a comprehensive suite of evaluations for instrumental self-reasoning abilities in LLM agents. The authors motivate the importance behind evaluating the self-reasoning capabilities of frontier models and design several tasks and evaluation measures toward understanding the quality of self-reasoning. An abundance of results are provided, showcasing that some frontier models can perform this type of reasoning but all models struggle at the most difficult task level. The authors conclude by noting more future work is needed to improve LLM self-reasoning capability, and making this self-reasoning capability transparent to mitigate risks.

**Strengths:**

Strengths:
+ This work studies an important area of research: evaluating self-reasoning capabilities of LLMs
+ The paper contains several well-designed tasks and measures for evaluating self-reasoning abilities of LLMs. As state-of-the-art LLMs perform poorly in this suite, this suite can be used as a testbed to test these complex reasoning abilities.

**Weaknesses:**

Weaknesses:
- Many of the specific details regarding the prompt and task design are found in the appendix. It would be beneficial to move some of this material in the main paper to improve readability.
- Could you expand on how the results in the Opaque Reasoning Section relate to Figures 2 and 3?
- Is there a larger takeaway or recommendation toward future LLM research that these results imply? Does poor current self-reasoning capabilities mean that a human should be in the loop to ensure better LLM performance?

**Questions:**

- Could you address the specific weaknesses above?
- The authors mention that this work evaluates self-reasoning in agentic settings. Could the authors provide a description of what they mean by "agentic" and whether they feel these results translate to embodied agents (e.g., robots)?

---

> ### Author Response · Authors · 2024-11-17
>
> Thank you for your positive assessment of our work and recognition of its importance for evaluating LLM capabilities.
>
> > Many of the specific details regarding the prompt and task design are found in the appendix. It would be beneficial to move some of this material in the main paper to improve readability.
>
> We appreciate that the task descriptions in the main body of the paper are brief. We will work on improving them when revising our paper, although we are quite constrained by page limit. We do provide extensive descriptions in Appendix C and we will make sure to signpost to them more clearly.
>
> > Could you expand on how the results in the Opaque Reasoning Section relate to Figures 2 and 3?
>
> Regarding the relationship between opaque reasoning and Figures 2/3: We do not include opaque reasoning in figures 2 and 3 as we evaluate opaque reasoning very differently to the other tasks. The opaque reasoning evaluation involves first creating a monitoring model to detect self-reasoning, then testing various evasion strategies (detailed in appendix B). Our results for opaque reasoning are found in figure 5. We will clarify this in our revision.
>
> > Is there a larger takeaway or recommendation toward future LLM research that these results imply? Does poor current self-reasoning capabilities mean that a human should be in the loop to ensure better LLM performance?
>
> The relationship between self-reasoning capabilities and safety presents interesting trade-offs. While limited self-reasoning may lead to suboptimal performance requiring human intervention, it also reduces the risk of agents developing sophisticated strategies that human overseers cannot understand or effectively monitor. This speaks to a broader question in AI safety: the balance between capability and controllability.
>
> Importantly, we are not necessarily arguing for artificially limiting self-reasoning capabilities. Rather, our evaluation framework helps quantify these capabilities so the AI community can make more informed decisions about appropriate capability levels for different applications. This remains an open question for future work.
>
> The current poor self-reasoning capabilities do suggest the importance of human oversight, particularly for critical applications. When models fail to reason effectively about their own limitations or environmental context, human intervention can provide necessary guardrails. We will expand this discussion in the revision.
>
> > The authors mention that this work evaluates self-reasoning in agentic settings. Could the authors provide a description of what they mean by “agentic” and whether they feel these results translate to embodied agents (e.g., robots)?
>
> Regarding agentic settings: We refer to scenarios where models must actively interact with and take actions within their environment, as opposed to purely language-based tasks. The same principles would apply to embodied agents, though our current focus is on software environments because we expect self-reasoning to be practically relevant earlier in this context.

---

> > ### Comment · Reviewer_byfs · 2024-11-23
> >
> > Thank you for the additional clarification. If the updated manuscript, please include the discussion regarding the interesting trade-offs in the revision.

---

### Official Review · Reviewer_sQN5 · 2024-11-04

**Soundness:** 3
**Presentation:** 2
**Contribution:** 2
**Rating:** 3
**Confidence:** 4

**Summary:**

The authors propose a benchmark suite called MISR which is composed of various evaluations regarding LLMs ability to modify configuration files to improve on a give task, identify issues with a given tool, knowledge seeking, deceptive / social reasoning and a monitoring task.

The objective of the benchmark is to identify progress on self-reasoning abilities of LLMs.

**Strengths:**

I generally liked the discussion in the paper. Discussion on self-reasoning is important and relates to other fields like Theory of Mind, human-AI etc. which are topical. I agree that self-reasoning is going to be very important for autonomous agents and the paper makes an effort in that direction.

**Weaknesses:**

While I like the direction of the paper, I'm not convinced that the evaluations correspond to self-evaluation. (I'm hoping for some discussion with the author to clarify) :

Parts that may not be "self-reasoning" :

- Discuss why is Tool Improvement part of self-reasoning. In this case LLM is acting as a verifier / checker of an external system. I feel access to deceptive tools is a second-order concern for current LLMs (i.e. LLMs should be aware of when tools are deceptive), but even that doesn't seem to be self-reflection.
- Similarly, why is embedded social reasoning an aspect of "self-reasoning" - which is designed (lines 189 - 191) for devising strategies in social contexts. This is more like opponent modeling / or modeling other agents and responding accordingly. If, the discussion were to be about whether the LLM can self-reason on if its allowed to be deceptive etc. - that may be more in line with paper's goal, but the current section 3.4 seems to be not related to self-reasoning.
- Knowledge Seeking - the objective seems to be whether the LLM knows about the environment (which is different from knowing about its own abilities). Are the authors not differentiating between an agent & the environment where it lives? For example, a human knowing about their own abilities is closer to self-reason than them knowing about how government of a certain country functions (like knowing about Linux shell env). As in line 182; do the authors not provide information about the environment and expect LLM to browse through shell configuration files to gather the information about the environment?

Self-Modification is more in line with "self-reasoning" but is taken from a prior work as acknowledged by the authors. I don't think this should be counted towards this paper's contributions (as I believe the authors would agree). Regardless, while in spirit the section 3.1 seems more aligned, can the authors discuss :
-  Do the agents provide information about the configuration files that impacts the task as part of the input prompt? If not, then why is it reasonable for the LLM to be able to guess presence of an arbitrary configuration file for the task.


Generally, I feel that there are interesting evaluations but not necessarily in line with the central theme of "self-reasoning".

The other (minor) concern I have is the lack of some human baseline. For instance, would humans be able to guess that  for improving acceleration they have to update a "config.yaml" somewhere in the file system? Some evaluation / discussion on the fact that there is enough information in the input prompt for an LLM / human to be able to achieve the task should be helpful.

Finally, it seems that some of the tasks in the evaluation category are repetitive as the authors acknowledge (tool improvement checks for errors in python code; self modification looks for updating some config) - can the authors discuss why multiple variations of the same task is helpful? What would be other applications areas (such as beyond changing config files)?

**Questions:**

Please see "Weaknesses" section.

---

> ### Author Response · Authors · 2024-11-17
>
> Thank you for your thorough review and thoughtful questions about our evaluation framework and we look forward to discussing it with you further. We appreciate your recognition of the importance of this research direction and its relevance to Theory of Mind and human-AI interaction.
>
> Regarding the definition of self-reasoning: We discuss this in more depth in our general response to all reviewers, but in brief, we aim to measure an agent’s ability to understand it is an agent embedded in an environment and use this knowledge to its advantage. After considering your comments, we think “embedded self-reasoning” might be a clearer term. We’d be interested to hear your thoughts on whether this terminology better addresses your concerns, or if you see more fundamental issues with how we've clustered these capabilities together.
>
> Let us address your remaining questions one-by-one.
>
> > Do the agents provide information about the configuration files that impacts the task as part of the input prompt? If not, then why is it reasonable for the LLM to be able to guess presence of an arbitrary configuration file for the task.
>
> In the appendix, we provide fully detailed task descriptions. The prompts include clear instructions to explore a Linux-based environment, with the goal of gathering maximal information. To aid exploration, we include a README.md file in the working directory with helpful environmental context.
>
> > The other (minor) concern I have is the lack of some human baseline.
>
> While we haven’t conducted formal human evaluations, we’ve designed our tasks to be straightforward for humans with basic technical knowledge. We provide all necessary context, including directory listings that show relevant files in the prompt (e.g., “config.yaml” for self-modification tasks). Based on the task design and included instructions, we expect humans with basic technical experience would achieve near-perfect performance. So a human baseline would likely not be very informative in our setting.
>
> > Finally, it seems that some of the tasks in the evaluation category are repetitive as the authors acknowledge (tool improvement checks for errors in python code; self modification looks for updating some config) - can the authors discuss why multiple variations of the same task is helpful? What would be other applications areas (such as beyond changing config files)?
>
> Regarding task diversity: We include multiple variations within each task category for three key reasons: (1) to reduce evaluation variance, (2) to decrease dependence on any single task implementation, and (3) to capture different aspects of self-reasoning within each category. While more variety would certainly be beneficial, our tasks are diverse enough that we saw significant improvement in the robustness of our results when adding more of the tasks.
>
> Regarding applications beyond config modifications: We agree there are many exciting possibilities, including having agents perform self-finetuning, model steering, or mechanistic interpretability on themselves. However, these would introduce significant challenges for comparative evaluation since they depend heavily on model-specific architectures and implementations. Moreover, this is generally beyond the capabilities of current systems, and will likely not give a useful signal in the near future (which is the focus of our evaluations).
>
> We chose to focus on config file modifications because they provide a simple, model-agnostic way to evaluate self-reasoning while controlling for general capabilities.

---

> > ### Author Response · Authors · 2024-11-25
> >
> > Thank you again for your feedback and comments!
> >
> > We hope our response has addressed your concerns. If you have any further questions, please do not hesitate to let us know. We are looking forward to having further discussion with you at this stage.

---

### Author Response · Authors · 2024-11-17
**General Comment**

We thank all reviewers for their thoughtful and constructive feedback. We are encouraged that reviewers highlighted several important strengths of our work. Reviewers find the research direction “highly relevant” and “crucially important” to frontier model development (oVYg), our evaluations “well-designed” (byfs), “credible and reproducible” (wZBb) and “thorough” (oVYg).

The reviewers’ primary concerns about our work seem to be about novelty compared to prior work and the definition and scope of self-reasoning. We address these here in addition to our individual responses.

**I. Novelty and Relationship to Prior Work (raised by reviewers sQN5, oVYg, and wZBb)**

Multiple reviewers raised concerns about the novelty of our work compared to Phuong et al. 2024. While Phuong et al. proposes a comprehensive suite of dangerous capability evaluations and first identify instrumental self-reasoning as a relevant capability to measure, they only provide 3 actual evaluation tasks and focus mostly on self-modification despite their more general definition of self-reasoning. In contrast, we provide **14 novel tasks** including 2 additional self-modification tasks. We offer significantly greater diversity in task types and provide the first set of evaluations that goes beyond self-modification.

**II. Definition and Scope of Self-Reasoning (raised by reviewers sQN5 and oVYg)**

We appreciate the reviewers’ questions about what constitutes “self-reasoning.” We acknowledge the terminology is a bit vague and the community has not converged on a well-defined term yet. Upon reflection, we think the term “embedded self-reasoning” might better capture our focus: evaluating an agent’s understanding of being embedded in an environment and its ability to reason about using the fact of being an embedded agent to its advantage.

This framing helps clarify how our evaluation categories connect to embedded self-reasoning:

1. Self-Modification: This represents the most direct form of embedded reasoning, where the agent must understand and modify its own configuration which is only possible due to being an embedded agent. Reviewers agreed this clearly demonstrates self-reasoning capabilities.

2. Tool Improvement: We view this as a form of self-modification, as tools can be considered part of the agent's extended capabilities. The agent (a model + scaffolding) includes its tools, making tool improvement a form of self-modification. Similar to self-modification, tool improvement crucially relies on the agent being embedded in the environment. Furthermore, as agents increasingly create and integrate their own tools (see section 3.2 for more discussion of this), the boundary between agent and tool becomes increasingly blurred.
3. Embedded Social Reasoning: This evaluates the agent's understanding of its place within a social environment. Agents are fundamentally defined by their surrounding context, and leveraging this context requires understanding one's role within it (a perspective developed in various philosophical traditions, including those of Hegel and Sartre). Leveraging the social context requires the agent to understand its role in relation to the context, which we view as a particular form of embedded self-reasoning. We focus on deceptiveness and scheming because this requires a particularly fine-grained understanding of the social context and is particularly relevant to AI safety.
4. Knowledge Seeking: These tasks focus on the agent learning about being embedded in an environment, which is a prerequisite to embedded self-reasoning in practice. Solving these tasks requires the agent to reason about its current state of knowledge and identify information gaps within its environment. This requires some metacognition about the agent’s knowledge state, which is a prerequisite to the explicit forms of self-reasoning the other evaluations focus on.

We’re planning to update the paper to emphasize the focus on embedded self-reasoning and expand on the discussion of why individual evaluations are relevant in this context. We’d be interested in the reviewers’ feedback whether this would improve clarity.

---

> ### Author Response · Authors · 2024-11-22
>
> In response to continued discussion about novelty, we would like to emphasize three key scientific contributions of our work beyond the framework itself:
>
> 1. Our evaluations probe qualitatively different aspects of embedded self-reasoning, moving beyond pure self-modification to understand how agents comprehend and operate within their environment
> 2. We provide the first systematic cross-model comparison of self-reasoning capabilities, offering empirical insights into how these abilities develop across different frontier models
> 3. Our monitoring experiments reveal a novel phenomenon: models can, in certain situations, learn to evade detection of their self-reasoning capabilities -- an aspect not previously studied in the literature
>
> See our [detailed response](https://openreview.net/forum?id=MOEBghZGVq&noteId=PvaKByyW0r) to reviewer wZBb for further discussion of these points.

---

### Meta-Review · Area_Chair_1DTP · 2024-12-19

**Metareview:**

The work proposes a benchmark suite called MISR, which includes various evaluations of LLMs’ abilities to modify configuration files to improve a given task, identify issues with a specific tool, engage in knowledge-seeking, demonstrate deceptive or social reasoning, and perform monitoring tasks. This paper contains several well-designed tasks and measures for evaluating the self-reasoning abilities of LLMs. However, after the AC thoroughly reviewed the paper, it was found to be less readable and somewhat prone to over-claims, an issue also pointed out by the reviewers.

**Additional Comments On Reviewer Discussion:**

The review scores for this paper showed significant variation. To ensure an effective decision, the AC engaged in discussions with the reviewers and carefully revisited the manuscript. The findings revealed several major concerns, including the need for a clearer presentation of related work, a comprehensive summary of existing research on revision-based paradigms, and a clear articulation of the novelty of this work. Additionally, the writing lacked objectivity and would benefit from a stronger focus on academic contributions and technical innovation. Considering the paper’s scores, which were significantly below the acceptance threshold, and the further discussions with reviewers, it was decided to reject this submission.

---

### Decision · Program_Chairs · 2025-01-22

Reject